# Development and Characterization of Thermal Water Gel Comprising *Helichrysum italicum* Essential Oil-Loaded Chitosan Nanoparticles for Skin Care

**Sofia M. Saraiva** [1], **Ana Margarida Crespo** [1], **Filipa Vaz** [1], **Melanie Filipe** [1], **Daniela Santos** [1], **Telma A. Jacinto** [1], **Ana Cláudia Paiva-Santos** [2,3], **Márcio Rodrigues** [1,4], **Maximiano P. Ribeiro** [1,4], **Paula Coutinho** [1,4] and **André R. T. S. Araujo** [1,5,*]

1   CPIRN-IPG, Center of Potential and Innovation of Natural Resources, Polytechnic Institute of Guarda, Av. Dr. Francisco de Sá Carneiro, No. 50, 6300-559 Guarda, Portugal
2   Department of Pharmaceutical Technology, Faculty of Pharmacy of the University of Coimbra, University of Coimbra, 3000-548 Coimbra, Portugal
3   REQUIMTE/LAQV, Group of Pharmaceutical Technology, Faculty of Pharmacy of the University of Coimbra, University of Coimbra, 3000-548 Coimbra, Portugal
4   CICS-UBI, Centro de Investigação em Ciências da Saúde, Universidade da Beira Interior, Av. Infante D. Henrique, 6200-506 Covilhã, Portugal
5   Department of Chemical Sciences, Laboratory of Applied Chemistry, Faculty of Pharmacy, LAQV, REQUIMTE, Porto University, Rua de Jorge Viterbo Ferreira, 228, 4050-313 Porto, Portugal
*   Correspondence: andrearaujo@ipg.pt

**Abstract:** *Helichrysum italicum* essential oil (*H. italicum* EO) is recognized for its anti-inflammatory, antimicrobial and wound-healing properties. The main goal of the present work was the development and characterization of a gel formulation comprising *H. italicum* EO loaded in chitosan nanoparticles (NPs) for dermatological applications. *H. italicum* EO-loaded chitosan NPs presented hydrodynamic diameter and PdI of about 300 nm and 0.28, respectively, and a surface charge of +19 mV. The *H. italicum* EO-loaded chitosan NPs were prepared by means of ionic gelation and then incorporated into a thermal water gel formulation. The organoleptic and physicochemical properties of the developed gel were studied. The gel remained stable under accelerated test conditions, maintaining pH, viscosity and organoleptic properties. In addition, the formulation presented pH, viscosity and spreadability properties suitable for topical application. Finally, the performance of the gel in topical application was evaluated on the skin of volunteers using non-invasive methods, particularly, by means of biometric evaluation. These assays showed that the properties of the developed thermal water-based gel formulation with *H. italicum* EO-loaded chitosan NPs can improve skin hydration and maintain healthy skin conditions, demonstrating its putative role for distinct dermatological applications.

**Keywords:** *Helichrysum italicum*; essential oil; chitosan nanoparticles; thermal water; gel; skin

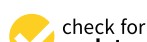



## 1. Introduction

*Helichrysum italicum* (Roth) G. Don (Asteraceae), also called "immortelle", is a small aromatic shrub with yellow flowers and is characteristic of the Mediterranean, blooming on sandy soil and cliffs [1,2]. The flowers, as well as other structures of *H. italicum* plant, have been commonly used in traditional medicine, as infusions, decoctions or ointments, for treating conditions affecting the skin (e.g., burns, wounds and psoriasis), and the respiratory and digestive systems due to their anti-inflammatory, antimicrobial and wound-healing properties [3]. As with other plants, extracts and essential oils (EOs) can be obtained from *H. italicum*. EOs are characterized as highly concentrated and volatile liquids. The main constituents of *H. italicum* EO are flavonoids, pyrones, terpenes, terpenoids, phloroglucinols and acetophenones [2,4]. These compounds are responsible for

the reported antimicrobial, antioxidant and anti-inflammatory activities [1,3,5], as well as anti-collagenase and anti-elastase properties, of *H. italicum* EO [6]. Due to these properties, *H. italicum* EO has been used for the management of dermatological conditions and for cosmetic applications [7]. Topical formulations containing *H. italicum* EO have also been proposed for treating diabetes mellitus-associated comorbidities (foot ulcers) [8] and for antiaging purposes [9]. *H. italicum* EO has been reported to be safe for topical use up to 0.5% (*w/w*) without causing irritation [8,10].

In order to improve EO potential, as a means for providing sustained release and enhancing its penetration through the tissue of interest, several EOs have been encapsulated in nanocarriers [11–16]. Furthermore, EO nanoencapsulation overcomes EOs' low solubility in water and decreases oxidation and photothermal degradation rates [11].

Among the different types of nanocarriers used for EO delivery, liposomes are a common and widely explored option, owing to their lipidic composition that provides occlusive properties and enables a higher interaction with skin components, improving skin penetration [17,18]. On the other hand, polymeric nanoparticles (NPs), namely, chitosan NPs, have also been shown to have great potential for EO delivery in applications such as antibacterial, anti-cancer, wound-healing and skin condition applications [17,19–22], among others. Chitosan is a biocompatible and biodegradable polysaccharide with cationic net charge. Chitosan/Sodium tripolyphosphate (TPP) NPs and pectin/chitosan NPs were recently used for the encapsulation of *Boswellia sacra* EO and jasmine EO, respectively, for anti-cancer applications [19,21]. Attallah et al. [19] showed that pectin/chitosan NPs were able to improve EOs stability, antioxidant potency and cytotoxic effects against breast cancer cells. On the other hand, chitosan NP loading with ginger EO [20], and turmeric and clove EOs [22] were explored for wound healing. Buntum et al. [22] demonstrated that the type of EO influenced the physicochemical properties of the nanosystems. Specifically, turmeric EO-loaded chitosan NPs presented sizes between 375 and 888 nm, while clove EO-loaded chitosan NPs presented smaller sizes (206 to 185 nm) with less variability, using the same chitosan:EO *w/w* ratios (1:0.25 to 1:1.0). Considering the different compositions of chitosan NPs found in the literature and the different EOs encapsulated, performing a direct comparison of NP properties is not feasible. Still, such studies demonstrated the suitability of these NPs for EO encapsulation for different applications. To the best of our knowledge, so far, there are no studies reporting the nanoencapsulation of *H. italicum* EO for dermatological applications. In order to enable easy application and further contribute to improve skin health, EO-loaded NPs were incorporated in gel mainly composed of Portuguese Cró thermal water with a unique composition. Cró thermal water is medium mineral water that is sulfurous water and also rich in silica and certain cations with important functions in the skin (K$^+$, Na$^+$ and Ca$^{2+}$) (Table 1) [23]. This composition is responsible for promoting cellular renewal, skin hydration, recovery of cutaneous barrier function and keratolytic effects. In addition, it has been demonstrated that this water possesses antimicrobial, antioxidant and anti-inflammatory activities [23–25].

**Table 1.** Cró thermal water composition. Adapted from [23].

| Physicochemical Composition (mg/L) | |
|---|---|
| Total sulfur (in I$_2$ 0.01 N) | 16.9 |
| Bicarbonate | 157 |
| Sulphate | 14.1 |
| Chloride | 33 |
| Fluoride | 15.7 |
| Silica | 47.8 |
| Sodium | 103 |
| Calcium | 3.5 |
| Magnesium | 0.21 |
| Potassium | 2.7 |

The potential of thermal water has been explored by different cosmetic companies, especially French ones, providing scientific evidence, through in vitro and in vivo studies, of the value of thermal water for the mentioned skin conditions, as recently reviewed [26,27].

The general objective of this work was the development and characterization of a thermal water-based gel formulation containing EO of *H. italicum* encapsulated in chitosan NPs for dermatological applications.

## 2. Materials and Methods

### 2.1. Reagents

Polysorbate 80, triethanolamine 99% and imidazolidinyl urea were acquired from Guinama (Valencia, Spain). TPP and chitosan (MW 100–300 KDa) were bought from Acros Organics (Geel, Belgium). Propylene Glycol, menthol and Carbopol 940 (Carbomer) were acquired from Acofarma (Terrassa, Spain). All chemicals used were of analytical grade. *H. italicum* EO was kindly provided by Planalto Dourado (Freixedas, Portugal). Thermal water was obtained from its natural source, Termas do Cró (Sabugal, Portugal). Acetic acid and ethanol were acquired from VWR International (Porto, Portugal) and Honeywell (Seelze, Germany), respectively.

### 2.2. Preparation and Characterization of H. italicum EO-Loaded Chitosan Nanoparticles

The chitosan NPs were prepared by means of ionotropic gelation [28]. A chitosan solution (1% *w/v*) was prepared in acetic acid (0.5%, *v/v*). The solution was centrifuged (4000 rpm, 30 min) and filtered using a 0.45 μm cellulose nitrate filter. Afterwards, a stock solution of chitosan/polysorbate was prepared by mixing 0.23 mL of polysorbate 80 and 20 mL of chitosan solution. The mixture was placed in a water bath at 45 °C for 2 h.

For the encapsulation of EO, 25 μL of EO was added to 5 mL of the chitosan/polysorbate 80 mixture and left stirring for 40 min (Magnetic hotplate stirrer; VMS-C4 Advanced; VWR) at room temperature (RT). The chitosan NPs loaded with EO were formed by adding TPP solution (0.2% *w/v*) into the chitosan/polysorbate/EO mixture under sonication (Branson sonifier; model SFX 150; equipped with a microtip) for 40 min in an ice-bath to avoid overheating the formulation. Then, the NPs were isolated and recovered by centrifuging portions of 1 mL in 2 mL Eppendorf tubes at 14,000 rpm for 10 min at RT. The pellets containing the NPs were resuspended in thermal water (9 mL) and further sonicated for 40 min. The hydrodynamic size and polydispersity index (PdI) of three different batches of EO-loaded chitosan NPs were determined using dynamic light scattering, and the surface charge (zeta potential) was determined with laser doppler anemometry using Zetasizer NAnoZS® (Malvern Instruments, Worcestershire, UK).

### 2.3. Preparation of Gel Formulations

The gel containing EO-loaded NPs (Gel NPs-EO) was prepared by dissolving Carbopol 940 (0.5%), imidazolidinyl urea (0.3%) and menthol (1.0%) in ethanol (2.0%) and then slowly adding, at RT and while continuously stirring, a mixture of thermal water and propylene glycol (7.0%). Afterwards, 9 mL of the previously prepared EO-loaded NPs was added to the gel while stirring. The gel was neutralized with triethanolamine (to pH = 5.5) and stirred until a homogeneous appearance was obtained. Finally, the gel was transferred into a glass container protected from light and stored until further use.

For comparison purposes, base gel (without EO and NPs) and gel containing OE were also prepared (Gel-EO, without chitosan NPs). Gel-EO was prepared as previously described, and EO (0.5% *v/w*) was added to the gel, at RT, before the neutralization step with triethanolamine; it was stored in a glass container protected from light.

All the gel batches were prepared to present a final weight of 30 g.

## 2.4. Characterization and Storage Stability of the Gel Formulations

### 2.4.1. Organoleptic Properties and pH

The organoleptic properties of the gel formulations were analyzed regarding appearance, color and odor. In addition, the gel formulations stored at 4, 25 and 40 °C were analyzed monthly. The pH of gel was measured using a potentiometer (Mettler Toledo, Schwerzenbach, Switzerland). Three different batches of each type of gel were analyzed.

### 2.4.2. Viscosity

The viscosity of gel was determined using Brookfield rheometer DV3T® (Brookfield Engineering Laboratories, Inc., Middleboro MA, USA) equipped with a cone-plate set (CPA-52Z spindle) A sample of 0.5 g of gel was placed in the rheometer plate, and the viscosity was recorded from 2.5 to 20 rpm at 32 °C (representing skin temperature), repeating the cycle five times.

### 2.4.3. Spreadability

The analysis of gel spreadability, on gel stored at 25 °C, was performed monthly using TA-XTplus Texture Analyser (Stable Micro Systems, Surrey, UK) with a TTC spreadability rig test. The method was set for a probe penetration depth of 15 mm, a speed of 3.0 mm/s, and maximum compression and tension of 30 and 5 Kg, respectively. The sample was placed in the female cone, avoiding the incorporation of air. The downward movement of the male cone compressed the sample, promoting its spreading between the surfaces of the two cones. The measurements were performed in triplicate.

### 2.4.4. Accelerated Stability Studies

According to ISO/TR 18811:2018, the stability of cosmetic products can be evaluated following different methods. Herein, preliminary stability tests were performed using two different methods, namely, (i) centrifugation and (ii) freeze–thaw cycles.

On the one hand, 10 g of each gel was centrifuged for 30 min at 3000 rpm, and the process was repeated six times, to determine the capacity of gel to resist centrifugal force. On the other hand, the stability of gel when submitted to cold temperatures was also determined. For that, 30 g of gel was subjected to −20 °C (24 h) followed by 24 h at 25 °C. The freeze/thaw cycles were repeated five times. Gel presenting sedimentation, phase separation or any other sign of instability was considered unstable, while that maintaining its properties was considered stable.

## 2.5. Evaluation of Cutaneous Biometry

Three female volunteers (21 to 25 years old) participated in the study providing their informed written consent. Before applying the gel, the volunteers rested for about 30 min with the forearm uncovered to ensure normal blood circulation and acclimatization. In addition, the volunteers were instructed to not apply any cosmetics or other products on the forearms before and during the test. The gel was topically applied in a delimited area of the forearm of the volunteers. Each volunteer received one type of gel, (i) base gel, (ii) gel containing *H. italicum* EO or (iii) gel with chitosan NPs loaded with *H. italicum* EO. The effects of gel on the cutaneous hydration degree and transepidermal water loss (TEWL) were determined using Multi Probe Adapter equipment (MPA® Courage-Khazaka, Koln, Germany) equipped with Corneometer® and Tewameter® probes, respectively. The referred parameters were evaluated immediately after application (T0), after 30 min (T30) and after 60 min (T60), without removing the applied gel. The measurements were performed in quintuplicate.

## 2.6. Statistical Analysis

Statistical analysis was performed using GraphPad Prism (Version 6.0 software) (GraphPad Software, San Diego, CA, USA). The results are presented as means ± standard deviations (SDs). Data were compared using two-way ANOVA followed by Tukey's

multiple comparison tests. A *p*-value lower than 0.05 ($p < 0.05$) was considered statistically significant.

## 3. Results and Discussion

Despite the potential of *H. italicum* EO, this powerful concentrate presents limitations, such as insolubility in water and volatility. Novel solutions based on the encapsulation of EOs in nanocarriers, such as polymeric NPs, have been explored, also showing the capacity to increase EO permeation through the skin and improve the treatment outcome [29]. In this sense, in the present study, chitosan NPs were prepared for the encapsulation of *H. italicum* EO. The NPs were further incorporated into a gel formulation to improve its applicability to the skin. The physicochemical properties of the NPs and gel were studied. In addition, the effects of gel containing EO-loaded chitosan NPs on skin hydration and TEWL were studied in healthy volunteers using non-invasive methods.

### 3.1. Preparation and Characterization of Gel Containing H. italicum EO Loaded-Chitosan Nanoparticles

The chitosan NPs, loaded with *H. italicum* EO, were prepared using the simple, straightforward and widely known ionotropic gelation method [28]. This method consists in the addition of TPP solution into chitosan solution. In order to enhance NP stability, a non-ionic surfactant was also incorporated in the chitosan solution. The polysorbate-stabilized chitosan NPs loaded with *H. italicum* EO presented mean hydrodynamic size and PdI of $316 \pm 14$ nm and 0.284, respectively. As expected, the NPs presented a positive surface charge of about +19 mV, resulting from the positively charged amine groups of chitosan.

In order to enable easy application on the skin, the chitosan NPs loaded with *H. italicum* EO were incorporated into gel. In a previous study, our group demonstrated that gel containing thermal water (Portuguese Cró thermal water) improved skin hydration, TEWL and relief [24]. Such properties were mainly attributed to the thermal water. Considering the promising results observed, herein, we prepared gel based on Cró thermal water for the incorporation of the chitosan NPs loaded with *H. italicum* EO. The combination of the beneficial properties of both EO and thermal water in a dermatological formulation was envisioned as a promising option for sensitive skin as well as for the management of inflammatory skin conditions. As a control, base gel (without EO and NPs) and gel containing free EO (without NPs) were also prepared.

The different ingredients and concentrations used for the preparation of gel are commonly found in cosmetic products, thereby ensuring its safe use. Aside from the importance of selecting adequate and safe compositions, other factors must be taken into consideration during the development process of a dermocosmetic product, such as (i) stability, (ii) organoleptic properties (appearance and odor), (iii) pH, (iv) rheological properties and (v) texture.

As shown in Figure 1, the organoleptic characteristics and pH of the prepared gel formulations, i.e., base gel, Gel-EO and Gel-NPs EO, were similar.

### 3.1.1. Formulation Physicochemical Properties under Different Storage Conditions

A detailed study of the gel physicochemical properties was performed in order to evaluate gel organoleptic properties, pH, rheological properties, texture, spreadability and the ability to maintain such properties under storage conditions (4, 25 and 40 °C) over a period of three months.

No signs of degradation or instability were observed. All the formulations maintained a homogeneous appearance, whitish color and menthol odor. The pH of the gel samples was also determined. pH is an essential parameter to be determined when carrying out stability studies of cosmetic formulations, as this must be compatible with the pH of the area of the body where the formulations are applied. Since the gel herein prepared is intended for topical application, the pH of the gel samples should be ideally within the range of skin pH, which varies between about 4.5 and 5.75. For this, triethanolamine, a

commonly used alkalizer agent, was used to adjust the pH of gel to be within the desired range [24,30]. In addition, this compound helps in the formation of gel itself when added to Carbopol [30].

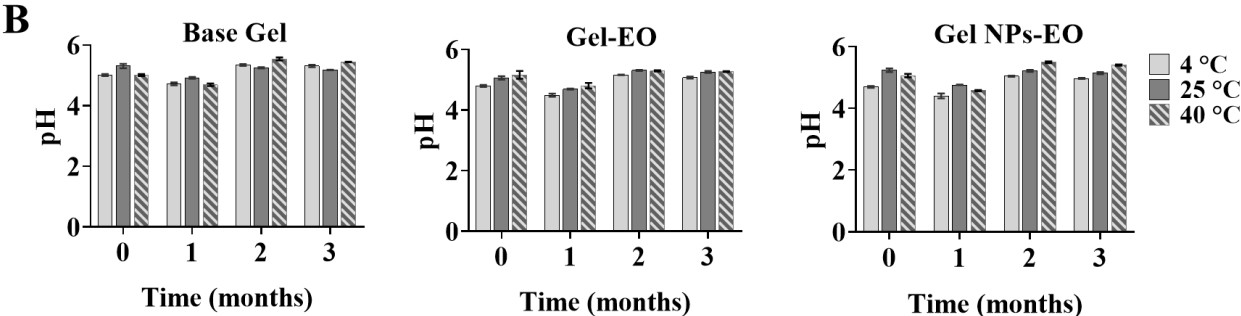

**A**

| Formulation | Organoleptic properties (T0) | 1 month (4 / 25 / 40 °C) | 2 months (4 / 25 / 40 °C) | 3 months (4 / 25 / 40 °C) |
|---|---|---|---|---|
| Base Gel | Homogeneous appearance; whitish color; menthol odor | ✓ | ✓ | ✓ |
| Gel-EO | | ✓ | ✓ | ✓ |
| Gel NPs-EO | | ✓ | ✓ | ✓ |

**B**

**Figure 1.** Evaluation of organoleptic properties (**A**) and pH (**B**) of gel samples (Base Gel, gel containing EO (Gel-EO) and gel containing EO-loaded chitosan nanoparticles (Gel NPs-EO)) at 4, 25 and 40 °C, over 3 months. Results are shown as the means ± SDs of 3 replicates.

As shown in Figure 1, the pH of gel with chitosan NPs loaded with *H. italicum* EO stored at 25 °C slightly decreased after one month but returned to the initial value and remained stable until the third month of the study. Regarding the samples stored at 4 °C and 40 °C, there were slight changes in pH, without compromising the suitability for skin topical application.

In short, the prepared gel samples, subjected to different conditions over three months, can be considered compliant since they remained within the pH range considered normal for the skin and no significant changes in the initial organoleptic properties were observed.

As previously discussed, the monitorization of the rheological properties of gel is a key parameter to take into consideration during the development of dermatological products. Aside from evaluating the pH and organoleptic properties, the impact of different storage conditions (4, 25 and 40 °C) on the viscosity of the gel samples was also studied over three months, as shown in Figure 2. The viscosity of base gel was maintained during the period of study, which, combined with the organoleptic and pH results, demonstrates the stability of the gel formulation. A similar behavior was observed in gel containing free EO. On the other hand, gel containing EO-loaded chitosan NPs presented a similar viscosity during the first two months and a slight increase in viscosity on the third month of storage at 25 °C and 40 °C. Still, there were no noticeable sensorial changes in the viscosity, texture nor spreadability of the gel formulation, indicating that its functionality was not compromised during storage.

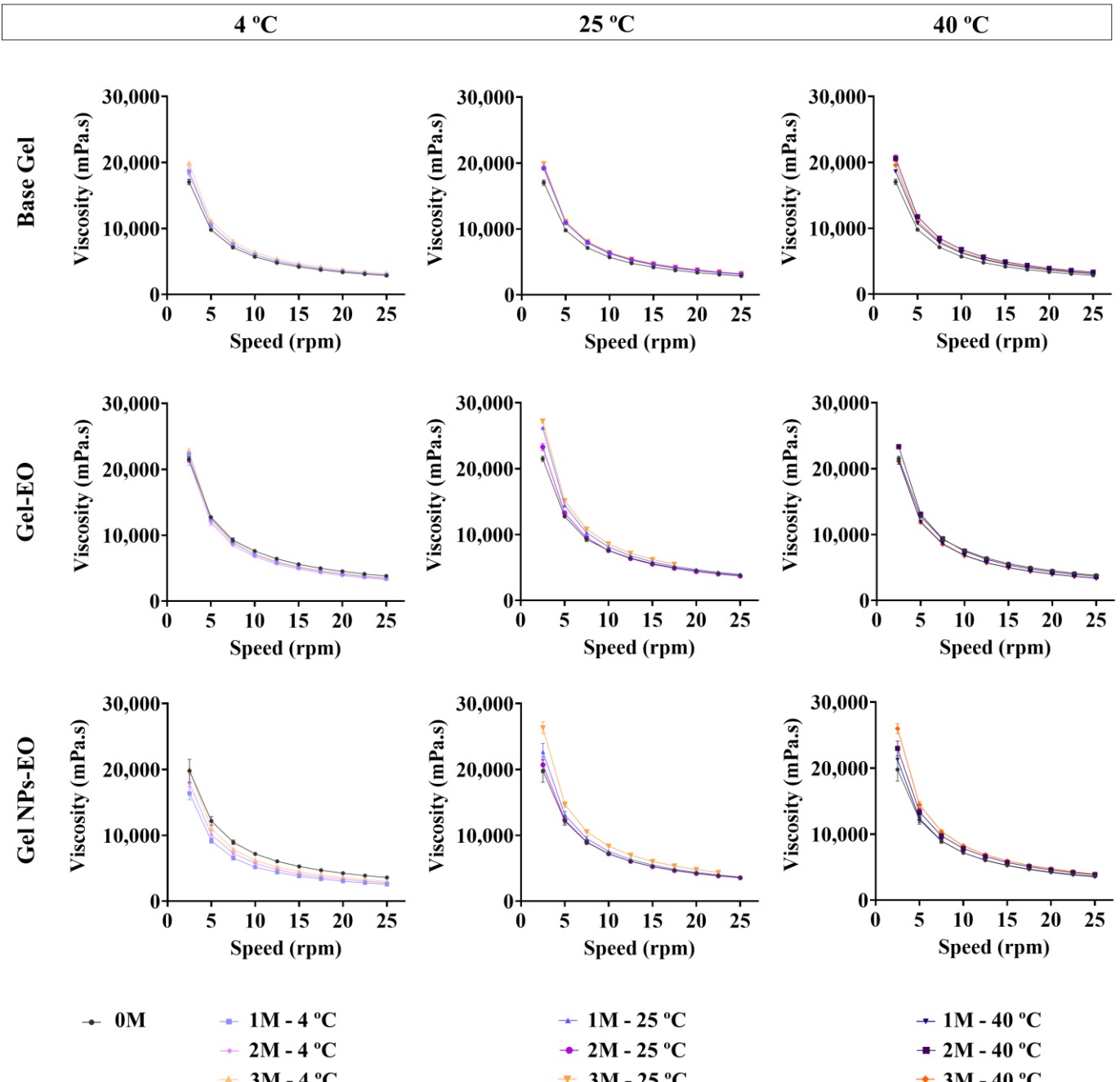

**Figure 2.** Evaluation of the viscosity (mPa·s) profiles of the prepared formulations: Base Gel, Gel-EO and Gel NPs-EO stored for 0, 1, 2 and 3 months (0 M, 1 M, 2 M and 3 M) at 4, 25 and 40 °C. Results are shown as the means ± SDs of 3 replicates.

### 3.1.2. Spreadability Analysis

According to the results of the spreadability analysis shown in Figure 3, base gel and gel containing EO were within a range that indicates their capacity to be easily spread over the skin and to adhere to the skin, providing sustained delivery of the bioactive principles. In addition, gel containing chitosan NPs presented the most consistent results along the three months of storage at 25 °C, probably indicating that the presence of chitosan played an important role in maintaining the gel properties over time. This could have been due to the electrostatic interaction of chitosan (positively charged) with Carbopol (polyacrylic acid, negatively charged), forming a more stable gel network [31,32].

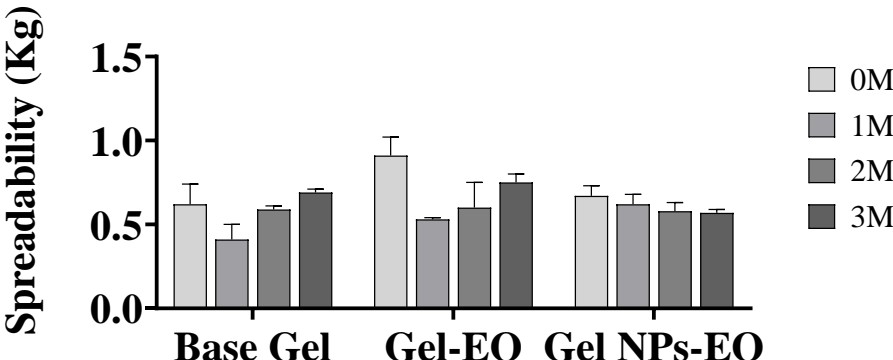

**Figure 3.** Spreadability analysis results of the prepared gel, i.e., Base Gel, gel containing free EO (Gel-EO) and gel with EO-loaded chitosan NPs (Gel NPs-EO), after preparation (0 M) and after 1, 2 and 3 months (M) of storage at 25 °C. Results are shown as the means ± SDs of 3 replicates.

3.1.3. Accelerated Stability Studies

Stability evaluation ensures that a product is stable during its lifetime. Stability is affected by factors such as light, temperature and humidity exposure. Thereby, stability studies aim at providing information on the physicochemical state of a product after it is exposed to conditions that can occur from manufacturing to the end of its shelf life.

Accelerated stability studies were performed using the centrifugation method and freeze/thaw cycles. The centrifugation method causes an increase in the movement of the particles present in the formulation, which can show an impact on the formulation stability and lead to precipitation, phase separation or coalescence [33]. The tested gel formulations did not show any signs of instability during or after the studies, demonstrating the stability of all the prepared gel samples (base gel, gel containing EO and gel containing chitosan NPs loaded with EO).

On the other hand, the freeze/thaw method consisted in the exposure of the formulations to extremely low temperatures (−20 °C) and then to RT (25 °C) and in repeating the process five times. This method tests the resistance of the formulations and enables the evaluation of the possible effects of storage temperature on product degradation, the formation of precipitates, turbidity or crystallization. After each freeze/thaw cycle, the organoleptic properties of all the tested gel formulations were evaluated. No differences were observed when compared to the initial (untested) product, thereby demonstrating once again that the formulations were stable. In addition, after five freeze/thaw cycles, the pH and viscosity of the formulations were also determined. As shown in Figure 4, after the freeze/thaw cycles, all the formulations presented a decrease in the pH mean value compared with the initial pH. Still, the pH of the formulations after the study was within the range of suitable pH for safe application on the skin and for avoiding any possible irritation.

Regarding the viscosity of the gel samples, the results showed that the freeze/thaw cycles did not compromise this characteristic, which remained identical to the initial viscosity (Figure 4). Viscosity allows the characterization of the rheological properties of a formulation to be performed, as it corresponds to the measurement of the resistance of a fluid to flow. The evaluation of this parameter makes it possible to assess whether the gel has adequate consistency/fluidity for easy topical application. In addition to the sensory impact, it also plays a pivotal role in the permeation of gel, since it is important that a formulation be easily applied to the thin layers of the skin.

3.1.4. Evaluation of Gel Formulation Efficacy with Cutaneous Biometry

The integrity of the skin, and its barrier capacity and protection effect against external factors can be compromised by different factors and can lead to the development of several dermatological conditions. Keeping the skin hydrated and avoiding the disruption of its hydrolipid barrier is essential [24,34]. Therefore, a careful design of dermatological

products designed for topical application is mandatory. As explained above, in this study, hydrophilic gel formulations composed of thermal water and containing *H. italicum* EO in free form as well as encapsulated in chitosan NPs, which aimed to overcome EO intrinsic drawbacks, were prepared.

**A**
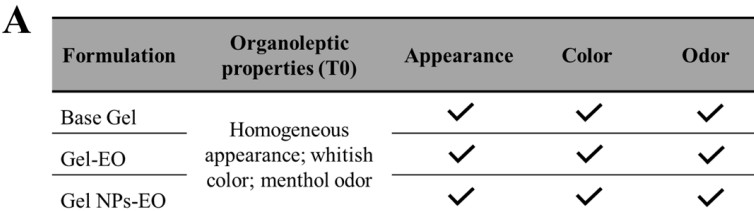

**B**
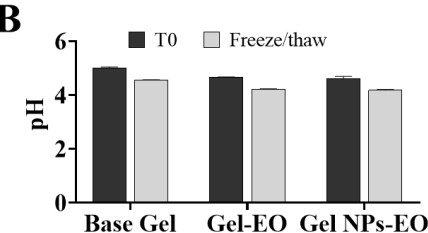

**C**
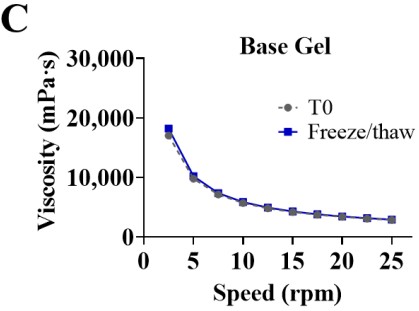
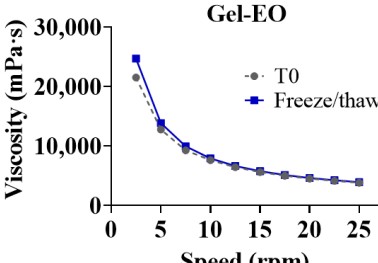
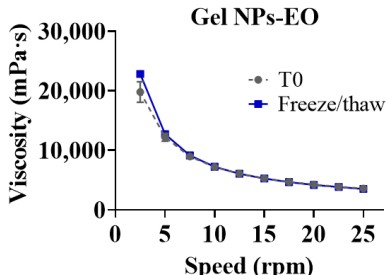

**Figure 4.** Evaluation of the organoleptic properties (**A**), pH (**B**) and viscosity (**C**) of the gels (Base Gel, gel containing free EO (Gel-EO) and gel containing EO-loaded chitosan nanoparticles (Gel NPs-EO)) after preparation (T0) and after five freeze/thaw cycles. Results are shown as the means ± standard deviations (SDs) of 3 replicates.

*H. italicum* EO was selected due to its anti-inflammatory, antimicrobial and antioxidant properties, which contribute to its beneficial properties for the management of wounds and skin conditions, among other disorders [2,3]. In addition, *H. italicum* EO was shown to have anti-collagenase and anti-elastase properties [6], which, combined with the above-mentioned properties, make this EO an interesting naturally derived material for therapeutic and cosmetic purposes. On the other hand, Cró thermal water was selected for the preparation of gel considering the previous study performed by our research group that showed its potential for the preparation of cosmetic formulations as well as the contribution to maintain the skin hydrated and healthy [24]. Similar effects were observed in other thermal water-based cosmetic products [26,27].

Therefore, after studying the stability of the prepared gel under different conditions and the impact on different relevant physicochemical properties, we performed a simple and preliminary evaluation of the efficacy of the prepared gel formulations through cutaneous biometry, a non-invasive technique.

The epidermal status of the volunteers' skin was assessed by determining essential parameters such as skin hydration and TEWL, according to the literature [35]. The volunteers' forearm cutaneous hydration and TEWL degrees were determined at different time points (0, 30 and 60 min after the application of the treatments), in a previously delimited area that received no treatment (non-treated) or one of the different gel formulations (base gel, Gel-EO and Gel NPs-EO). The monitoring of the skin status was performed using Corneometer® and Tewameter® probe determination at different time points after the application of gel.

As can be seen in Figure 5, non-treated skin presented consistent values of hydration and TEWL during the period of the study. At 0 min, volunteers' non-treated skin presented a hydration level of 51.5 ± 6.0 (a.u.) and TEWL values of 10.3 ± 0.2 $g \cdot h^{-1} \cdot m^{-2}$, which indicate that the volunteers had hydrated and healthy skin.

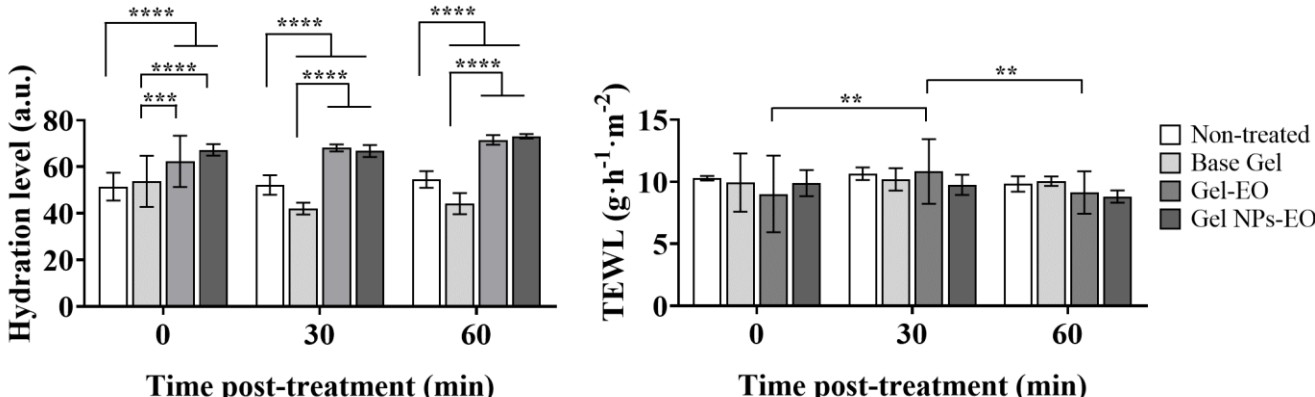

**Figure 5.** Effect of gel formulations (non-treated), Base Gel, gel containing free EO (Gel-EO) and gel containing EO loaded-NPs (Gel NPs-EO)) on hydration (arbitrary units; a.u.) and transepithelial water loss (TEWL, $g \cdot h^{-1} \cdot m^{-2}$) levels at 0, 30 and 60 min post-application in a delimited area of the volunteers' forearm skin. Three volunteers participated in the study. Results are shown as the means $\pm$ SDs of 5 measurements for each treatment and at every time point under study, ** $p < 0.01$, *** $p < 0.001$, **** $p < 0.0001$.

Immediately after the application of the treatments (0 min), the hydration level produced by base gel was similar to that of non-treated skin (not significant). On the other hand, both Gel-EO and Gel NPs-EO produced a significant increase in the skin hydration level when compared with non-treated skin ($p < 0.0001$). In addition, Gel NPs-EO produced a more significant increase ($p < 0.0001$) in skin hydration at 0 min than Gel-EO ($p < 0.001$) when compared with the base gel treatment.

After 30 and 60 min, the different gel formulations containing EO or NPs loaded with EO also led to a significant increase in the hydration levels compared with non-treated skin or base gel-treated skin. On the other hand, base gel caused a significant decrease in the skin hydration (vs. non-treated skin, $p < 0.0001$), demonstrating that the incorporation of *H. italicum* EO in the gel formulations either in free form or encapsulated in chitosan NPs was responsible for the significant beneficial effects herein observed in terms of skin hydration, which lasted for at least 60 min.

The skin ability to prevent water loss is directly related to stratum corneum (SC) barrier integrity. Dermatological products can improve skin hydration, as observed for Gel-EO and Gel NPs-EO, and further potentiate its effect by decreasing TEWL. Therefore, the effect of the treatments on skin TEWL was evaluated at the same time points. However, as depicted in Figure 5, no significant differences were observed among the types of treatment at 0, 30 or 60 min. On the other hand, when studying the effect of each treatment over time, Gel-EO showed to increase TEWL after 30 min (vs. Gel-EO at 0 min, $p < 0.01$), but after 60 min, no significant differences (vs. Gel-EO at 0 min) were observed. Regarding Gel NPs-EO, the slight TEWL level decrease observed 60 min after application on the skin (vs. non-treated at 60 min) was not statistically significant. This modest effect might be attributed to the presence of chitosan, which might have formed a layer on treated skin due to its positive charge and bioadhesive properties [30,36,37]. In general, none of the treatments significantly impacted TEWL over the 60 min of the study, indicating that, as expected, the selected composition did not compromise the hydrolipid barrier of the skin.

Overall, the gel formulations containing free EO and nanoencapsulated EO demonstrated to have potential to hydrate the skin and contribute to maintaining its healthy condition.

## 4. Conclusions

This preliminary work shows the combination of *H. italicum* EO and Portuguese Cró thermal water for dermatological applications. The gel formulation comprising *H. italicum* EO-loaded chitosan NPs demonstrated suitable properties for topical application, namely, organoleptic properties, pH, viscosity and rheological properties. Gel NPs-EO demon-

strated to be stable under accelerated stability conditions and for two to three months under different storage conditions. Skin hydration capacity, as one of the most relevant bioindicators of skin health, was evaluated using cutaneous biometry. The results confirmed that the hydrophilic gel containing EO-loaded chitosan NPs was able to improve skin hydration for at least 60 min post-application. Overall, Gel NPs-EO demonstrated potential for cosmetic applications. In the future, efficacy evaluation should be performed on a wider group of volunteers and ultimately on volunteers presenting inflammatory skin conditions to determine its therapeutic potential for the management of dermatological conditions.

**Author Contributions:** Conceptualization, M.R., M.P.R., P.C. and A.R.T.S.A.; methodology, investigation and formal analysis, A.M.C., F.V., M.F. and D.S.; writing—original draft preparation, S.M.S., T.A.J. and A.M.C.; writing—review and editing, S.M.S., A.C.P.-S., M.R., M.P.R., P.C. and A.R.T.S.A.; supervision, M.R., M.P.R., P.C. and A.R.T.S.A.; funding acquisition, M.R., M.P.R., P.C. and A.R.T.S.A. All authors have read and agreed to the published version of the manuscript.

**Funding:** This research was funded by Fundação para a Ciência e Tecnologia (FCT), Fundo Europeu de Desenvolvimento Regional (FEDER) and COMPETE 2020 financial support under the research project "The development of dermo-biotechnological applications using natural resources in the Beira and Serra da Estrela regions—DermoBio" (SAICT-POL/23925/2016). This research was also funded by Programa Operacional Regional do Centro (CEN-TRO-04-3559-FSE-000162) within the European Social Fund (ESF).

**Institutional Review Board Statement:** Informed consent was obtained from the enrolled volunteers, and all procedures were performed per the Helsinki Declaration. Institutional approval was not required to publish the data obtained from cutaneous biometry.

**Informed Consent Statement:** Informed consent was obtained from all subjects involved in the study.

**Data Availability Statement:** Not applicable.

**Conflicts of Interest:** The authors declare no conflict of interest.

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
