# Peer review of "Development and Characterization of Thermal Water Gel Comprising Helichrysum italicum Essential Oil-Loaded Chitosan Nanoparticles for Skin Care"

_cosmetics, doi:10.3390/cosmetics10010008_

Round 1
Reviewer 1 Report
I believe this article is ready for publication after proofreading of English language.
Methods are well described, figures well present results, references are up to date and I believe this study adds to the body of literature in this field.
Author Response
We thank the reviewer for the comments and the positive feedback. We performed an extensive revision of the English language. The corrections were performed using the “Track Changes” function of MS Word.
Reviewer 2 Report
Dear authors,
Thank you for the opportunity to see your interesting work.I have no objections to your experiments and analyses, but I am asking you to elaborate on some important issues.
In particular, the methodology is well-written but lacks many critical details.
Thermal water from the Termas do Cro – can you present the mineral concentrations in these waters?
At which temperature the “thermal waters” were added into the other component of the gel )(line 120 of your paper)
pH measurement of gel – which pH electrode have you used for gel pH verification (model, shape, etc.)
Viscosity – which geometry have you used – plate -plate or cone-plate or other – with your rheometer
Centrifugation – how big sample you used in your centrifuge – 1ml / 10ml /30ml or other? What was the size of a centrifuge cell?
Line 110 – “recovered by cenrifugation at 14000 g for 10 min ….. “ what it means 14000 g – I think it should be 14000 rpm for 10 min ? Am I right?
Line 106 – preparation of particles – mixture was homogenized for 40 min under stirring – What was the method of homogenization – only stirring or you use other homogenizer? What was the speed (velocity) of the stirrer? Or speed of homogenizer knife?
Line 208 – other studies …. Please, add references for this information. Are there a ref. 19 and 21?
I suggest moving the part of paragraph – from line 208 to line 221 from the Results into Introduction
Please, add into the fig. 2 name of the samples instead of A / B / C. In every other figure, the names are entered and it makes the work easier to read.
Line 111 the pellet containing of NP were resuspended in thermal water …. ? What size (mass or volume) of pellets was resuspended in thermal water ? and in which volume of thermal water? Can you estimate the concentration of NP in water, and later – can you estimate the NM concentration in the emulsion?
Author Response
Dear authors,
Thank you for the opportunity to see your interesting work. I have no objections to your experiments and analyses, but I am asking you to elaborate on some important issues.
In particular, the methodology is well-written but lacks many critical details.
Authors’ response: We thank the reviewer for the positive feedback and the constructive remarks. The missing details in the methodology section were addressed.
Thermal water from the Termas do Cro – can you present the mineral concentrations in these waters?
Authors’ response: Our group has previously reviewed the composition of thermal waters, including thermal water from Cró. The composition of Cró thermal water is presented below. We have added the reference of this work to the revised manuscript (introduction, reference 23) and we made the table.
Table 1. Cró thermal water composition. Adapted from Araujo et al. 2017.
|
Physicochemical composition (mg/L) |
|
|
Total sulfur (in I2 0.01 N) |
16.9 |
|
Bicarbonate |
157 |
|
Sulphate |
14.1 |
|
Chloride |
33 |
|
Fluoride |
15.7 |
|
Silica |
47.8 |
|
Sodium |
103 |
|
Calcium |
3.5 |
|
Magnesium |
0.21 |
|
Potassium |
2.7 |
At which temperature the “thermal waters” were added into the other component of the gel )(line 120 of your paper)
Authors’ response: The thermal water was added into the other components of the gel at room temperature.
pH measurement of gel – which pH electrode have you used for gel pH verification (model, shape, etc.)
Authors’ response: The pH was measured using a standard potentiometer (Mettler Toledo, Schwerzenbach, Switzerland).
Viscosity – which geometry have you used – plate -plate or cone-plate or other – with your rheometer
Authors’ response: A cone/plate set was used in the rheometer to perform the viscosity studies.
Centrifugation – how big sample you used in your centrifuge – 1ml / 10ml /30ml or other? What was the size of a centrifuge cell?
Authors’ response: After homogenization portions of 1 mL of nanoparticles were centrifuged in 2 mL Eppendorf tubes.
Line 110 – “recovered by cenrifugation at 14000 g for 10 min ….. “ what it means 14000 g – I think it should be 14000 rpm for 10 min ? Am I right?
Authors’ response: There was a typing mistake. The nanoparticles were recovered by centrifugation at 14000 rpm.
Line 106 – preparation of particles – mixture was homogenized for 40 min under stirring – What was the method of homogenization – only stirring or you use other homogenizer? What was the speed (velocity) of the stirrer? Or speed of homogenizer knife?
Authors’ response: The mixture was stirred for 40 min in order to homogenize it (Magnetic hotplate stirrer, VMS-C4 Advanced, VWR)
Line 208 – other studies …. Please, add references for this information. Are there a ref. 19 and 21?
Authors’ response: We updated the references of this paragraph, according to the reviewer comment.
I suggest moving the part of paragraph – from line 208 to line 221 from the Results into Introduction
Authors’ response: We followed the suggestion of the reviewer and moved the indicated paragraph to the introduction section.
Please, add into the fig. 2 name of the samples instead of A / B / C. In every other figure, the names are entered and it makes the work easier to read.
Authors’ response: We added the identification of each type of formulation to the fig. 2 and updated the figure in the revised manuscript.
Line 111 the pellet containing of NP were resuspended in thermal water …. ? What size (mass or volume) of pellets was resuspended in thermal water ? and in which volume of thermal water? Can you estimate the concentration of NP in water, and later – can you estimate the NM concentration in the emulsion?
Authors’ response: After recovery the NPs the pellets contained were concentrated and resuspended in 9 mL of thermal water (2.2). Then the EO-loaded NPs solution was added to the gel (2.3) to prepare a gel formulation with a final weight of 30 g. Consequently, all NPs produced were incorporated in one gel formulation, the process was replicated for each gel formulation (30 g). Despite we did not measure the concentration of NP, according to Hasheminejad, N. et al, (2019) it is estimated the yield of 32%, taking into account the approximate ratio 1:0.50 of (chitosan/polysorbate)/EO used (sum of dry weight of initial materials). Thus, it is estimated that the concentration of NP in thermal water was 2,66 mg/mL (w/v) and consequently it is estimated 23,94 mg/ gel formulation. Nevertheless, it would be interesting to determine the concentration of the nanoparticles. This could be performed by nanoparticle tracking analysis (NTA), however, our research facility and our collaborators do not have this equipment. We will keep this in mind for future studies.
The modifications made in the manuscript were performed with using the “Track Changes” function of MS Word.
References
Araujo, A.R.T.S.; Sarraguça, M.C.; Ribeiro, M.P.; Coutinho, P. Physicochemical fingerprinting of thermal waters of Beira Interior region of Portugal. Environ. Geochem. Health 2017, 39, 483–496, doi:10.1007/s10653-016-9829-x.
Hasheminejad, N., Khodaiyan, F., Safari, M. Improving the antifungal activity of clove essential oil encapsulated by chitosan nanoparticles. Food chemistry 2019, 275, 113-122. doi:10.1016/j.foodchem.2018.09.085